# A Highly Accurate Method for Deformation Reconstruction of Smart Deformable Structures Based on Flexible Strain Sensors

**DOI:** 10.3390/mi13060910

**Published:** 2022-06-08

**Authors:** Chengguo Yu, Xinyu Gao, Wenlin Liao, Zhili Zhang, Guishan Wang

**Affiliations:** 1Xi’an Research Institute of High Technology, Xi’an 710025, China; yuchengguo@cardc.cn; 2Facility Design and Instrumentation Institute, China Aerodynamics Research and Development Center, Mianyang 621000, China; gaoxinyu@cardc.cn (X.G.); liaowenlin@cardc.cn (W.L.)

**Keywords:** smart deformable structures, large deflection, deformation reconstruction, strain-moment relationship, elliptic integral, flexible strain sensors

## Abstract

Smart deformable structures that integrate designing, sensing, and controlling technology have been widely applied in the fields of aerospace, robotics, and biomedical engineering due to their multi-functional requirements. The deformation reconstruction method essential for security monitoring and shape controlling, especially for the large deflection deformation, remains a challenge on accuracy and efficiency. This paper takes a wind tunnel’s fixed-flexible nozzle (FFN) plate as the research object to develop a highly accurate deformation reconstruction method based on sensing information from flexible strain sensors. The mechanical behaviors of the FFN plate with large deflection deformation, which is modeled as a cantilever beam, are studied to analyze the relationship of the strain and moment. Furthermore, the large deflection factor and shell bending theory are creatively utilized to derive and modify the strain–moment based reconstruction method (SMRM), where the contour of the FFN plate is solved by particular elliptic integrals. As a result, structural simulation based on ABAQUS further demonstrates that the reconstruction error of SMRM is 21.13% less than that of the classic Ko-based reconstruction method (KORM). An FFN prototype accompanied by customized flexible sensors is developed to evaluate the accuracy and efficiency of the SMRM, resulting in a maximum relative error of 3.97% that is acceptable for practical applications in smart deformable structures, not limited to the FFN plate.

## 1. Introduction

Smart deformable structures are broad categories that indicate the working conditions when multi-functions are required. The FFN plate chosen as the research subject is the core structure of a wind tunnel to provide multiple Mach numbers of airflow. Specifically, the advanced wind tunnels have been widely designed for the independent research and development of aircraft, which motivates the development of aerodynamics and the related disciplines [1,2,3,4]. In the supersonic wind tunnel, the FFN that owns the deformable ability is an essential aspect of obtaining uniform airflow under the designed Mach number, as the quality of the FFN contour closely determines the flow quality of the wind tunnel [5,6]. In general, the FFN contour can be reconstructed by a laser tracker on the occasion of offline adjustment [7]. Nevertheless, it is difficult to obtain the real-time FFN contour in the flow calibration and shape control occasions due to the installation difficulty of the laser system, which affects the debugging efficiency and control safety of the FFN. The sensing network based on strain sensors behind the flexible plate provides an available approach to settle the above problem. However, the associated reconstruction method with superior accuracy and efficiency, crucial for structural monitoring and safety control, remains a challenge.

In some supersonic wind tunnels, dedicated protective structures were designed to limit the maximum curvature of the FFN plates for safety operation [8,9]. However, extra structures bring difficulties to the manufacturing, assembling, as well as maintenance. Strain sensors, such as strain gauges [10,11] and optical fiber sensors [12,13], were considered another proper way for structural health monitoring and safety operation. Nie et al. achieved the theoretic relationship between the displacement and the bending stress of the FFN plate by mechanical dynamics simulation [14]. Gao et al. introduced strain gauges to the intensive stress location of the FFN plate in case the stress surpasses the FFN plate’s strength caused by the synchronized movement of actuators [11]. Similarly, Ma et al. integrated threshold stress detection and synchronicity monitoring into the control system to guarantee structural health [15]. On the whole, strain sensors presently applied on FFN are only limited to the local stress monitoring and are still far from the overall deformation reconstruction that can provide more information of the FFN.

A large amount of research has been conducted on the deformation reconstruction method of flexible beams. William et al. proposed the Ko theoretic method for deformation reconstruction, which was verified in beams with distinctive sections [16,17]. After that, Ding et al. studied the deformation reconstruction of carbon fiber laminates based on Ko theory [18]. Liu et al. put forward a kind of deformation measurement method of machine structure using Ko theoretic method [19]). Ko theory–based methods, however, derived from the hypotheses of small deflection and changeless axis directions do not have enough capacity to reconstruct large deflection deformation. Tessler et al. from Langley Research Center (NASA) introduced the least square variations equations to develop an inverse finite element method (IFEM) for reconstruction problems [20]. Song et al. proposed a new method to reconstruct the dynamic deformation of beam structures by using fiber Bragg grating sensors and iFEM [21]. Zhao et al. proposed a real-time monitoring method of the wing state using the inverse finite element method. You et al. developed a novel two-node inverse beam element for two-dimensional deformation estimation of beam-like structures [22]. The accuracy of the above iFEM-based methods heavily depends on the classification of cells and the number of nodes, resulting in complicated structures and low efficiency.

This work focuses on developing a highly accurate reconstruction method for large deflection deformation of smart deformable structures from which the FFN plate has taken as the subject. First, the FFN plate is simplified as a beam for building a mechanical model. The strain–moment relationship of the FFN is analyzed and introduced to the mechanical model, followed by the derivation of differential equations of the FFN plate contours. As a result, the elliptic integral expression is applied to the differential equations to obtain the analytical solution. The sheet deformation theory further revises the reconstruction solution to reduce reconstruction errors. For data acquisition, flexible strain sensors developed in our previous paper have been installed on the FFN plate [23,24,25]. Experimental results demonstrate that the revised deformation reconstruction method based on the strain–moment relationship (SMRM) highly improves the reconstruction accuracy to 21.13%, which could be utilized for the structural health monitoring and deformation reconstruction of the FFN plate, morphing wings, and other smart deformable structures.

## 2. The Mechanical Model of the FFN Plate with Multiple Pivots

The FFN plate is among the most practical flexible structures that demand deformation reconstruction and health monitoring. As depicted in Figure 1a, the FFN can be divided into three primary sections, named the subsonic section, the throat section, and the ultrasonic section, according to the velocity of the inner flow. The Mach number of the flow velocity in these sections increases from lower than 1 to higher than 1 due to the shape change of the cross-section. The final Mach number in the wind tunnel is determined by the contour of the FFN plate that is driven by several controllable jacks (Figure 1b). Figure 1b illustrates that one side of the FFN plate is pinned to the wall of the wind tunnel while the other side is clamped to the part of the throat section, resulting in this specific deformation problem with constraints. Note that only the green part in Figure 1 represents the flexible shell where the bending modulus is applied. The movement of the throat section in black color (Figure 1) is determined by the terminal position of the above flexible shell after its forced deformation, which means that the deformation of the flexible shell and the movement of the throat section are correlated in this mechanical model.

By knowing the deformation mechanism, the mechanical model of the FFN plate could be built starting with force analysis shown in Figure 1b. The coordinate system is set on the fixed point of the FFN plate on which the M1∼M3 and the S1∼S6 denote the modulus and strain sensors, respectively. Note that the thickness of the FFN plate could be ignored compared with the scale of the width and length, leading to the hypothesis that the FFN plate can be equivalent to a flexible shell. Besides this, the effects of warpage and transverse shear are negligible as the forced displacement of the flexible shell is uniform along with the width direction. Thus, the mechanical model that aims at obtaining the contour of the FFN plate will be built equivalent to the Euler beam, calculating the analytical solution with the deformation theory of beams and revising the solution with the strain–moment relationship of the flexible shell. Figure 2a further illustrates the mechanical model that simplifies the FFN plate as a beam, which provides an easier way to obtain the analytical solution of the FFN plate’s contour.

## 3. The Derivation and Modification of the Reconstruction Algorithm

### 3.1. Relationship of the Strain and Moment

According to the geometry equation of Euler beam, the strain ϵ and the curvature ρ of a flexible beam can be described as Equation (Equation 1).
(1)$$\epsilon \left(x\right)=-\frac{h}{2\rho (x)}$$

When in-plane bending happens to the beam, the relationship of the curvature ρ with the bending moment *M* can be given by the Hook’s Law
(2)$$\frac{d\theta }{ds}=\frac{1}{\rho (x)}=-\frac{M(x)}{EI}$$
where θ and *s* represent the bending angle and arc length of the beam (Figure 3), respectively.

Then comes the relationship of bending moment by combing Equation (Equation 1) with Equation (Equation 2) as
(3)$$M\left(x\right)=\frac{2EI\epsilon (x)}{h}$$
where *h*, *E*, and *I* denote the thickness, Young’s modulus, and the inertial moment of the flexible shell, respectively. *x* represents the coordinate location of the beam.

### 3.2. Derivation of the Deformation with Large Deflection

For the ith segment of the beam, there is
(4)$$k_{i}=\frac{{\hat{\epsilon }}_{2i-1}-{\hat{\epsilon }}_{2i}}{x_{2i-1}-x_{2i}}$$
where ${\hat{\epsilon }}_{i}$ denotes the strain data from the sensor installed on the ith segment as shown in Figure 2b. $k_{i}$ is the coefficient that can be applied to interpolate the strain of every *x* from the finite data of the strain sensors. Thus, derived from Equation (Equation 2), there is
(5)$$M_{i}\left(x\right)=\frac{2EI(k_{i}\left(x-x_{2i}\right)+{\hat{\epsilon }}_{2i})}{h}$$

Then, the differential of Equation (Equation 2) can be written as
(6)$$\frac{d}{ds}\left(\frac{d\theta }{ds}\right)=-\frac{d}{ds}\left(\frac{M(x)}{EI}\right)$$
(7)$$\frac{d}{ds}\left(\frac{d\theta }{ds}\right)=-\frac{2k_{i}}{h}\cos\theta$$
dθ times both sides of Equation (Equation 7), resulting in
(8)$$\frac{d\theta }{ds}d\left(\frac{d\theta }{ds}\right)=-\frac{2k_{i}}{h}\cos\theta d\theta$$

After integration, the above equation can be written as
(9)$$\frac{1}{2}(\frac{d\theta }{ds}{)}^{2}=-\frac{2k_{i}}{h}\sin\theta +C_{i}$$

For the first segment $l_{1}$, the boundary conditions are
(10)$$\frac{d\theta }{ds}=\frac{M_{1}\left(x\right)}{EI}=\frac{2}{h}\left[k\left(l_{1}-x_{2}\right)+{\hat{\epsilon }}_{2}\right],\;\theta =\theta_{1}$$

Taking Equation (Equation 10) into Equation (Equation 9), $C_{1}$ can be solved as
(11)$$C_{1}=\frac{2{\left[k_{1}\left(l_{1}-x_{2}\right)+{\hat{\epsilon }}_{2}\right]}^{2}}{h^{2}}+\frac{2k_{1}}{h}\sin\theta_{1}$$

Then, we apply definite integral to Equation (Equation 9), with θ from 0∼$\theta_{1}$ and *s* from 0∼*h*, resulting in
(12)$$\int_{0}^{\theta_{1}}\frac{1}{\sqrt[{}]{2(-\frac{2k_{1}}{h}\sin\theta +C)}}d\theta =\int_{0}^{l_{1}}ds$$
set
(13)$$\alpha_{1}=\sqrt{\frac{k_{1}}{EI}},\;p_{1}=\frac{M_{1}{\left(x\right)}^{2}}{2k_{1}EI}.$$

The elliptic integral of the above equation can be calculated by
(14)$$l_{1}=\frac{\sqrt{2}}{\alpha_{1}\lambda }f$$
where $\lambda =\sqrt{\sin\theta_{1}-p_{1}-1}$, $f=F\left(\gamma_{1},t\right)-F\left(\gamma_{2},t\right)$. F(.) is the incomplete elliptic integral of the first kind, which is given as
$$\begin{array}{cc} \gamma_{1} & =\sqrt{\frac{p_{1}}{-\sin\theta_{1}+p_{1}-1}}\\ \gamma_{2} & =\sqrt{\frac{\sin\theta_{1}-p_{1}}{\sin\theta_{1}-p_{1}+1}}\\ t & =\sqrt{\frac{\sin\theta_{1}-p_{1}+1}{\sin\theta_{1}-p_{1}-1}} \end{array}$$

The bending angle $\theta_{1}$ of the first segment can be achieved by solving Equation (Equation 14). For any point on the first segment, the bending angles definitely belong to $\left[0,\theta_{1}\right]$ and their coordinates can be calculated by
(15)$$\begin{array}{cc} x & =\frac{\sqrt{2}}{\alpha }\left(\sqrt{p_{1}-\sin\theta_{1}}+\sqrt{\sin\theta +p_{1}-sin\theta_{1}}\right)\\ y & =\frac{1}{\sqrt{2}\alpha }\frac{1}{\lambda }\left[f^{\prime }+\left(\sin\theta_{1}-p_{1}-1\right)e^{\prime }\right)] \end{array}$$
as
$$\begin{array}{cc} f^{\prime } & =F\left(\gamma_{1}^{\prime },t\right)-F\left(\gamma_{2},t\right)\\ e^{\prime } & =E\left(\gamma_{1}^{\prime },t\right)-E\left(\gamma_{2},t\right)\\ \gamma_{1}^{\prime } & =\sqrt{\frac{\sin\theta +p_{1}-\sin\theta_{1}}{p_{1}-\sin\theta_{1}-1}} \end{array}$$
where E(.) is the incomplete elliptic integral of the second kind.

The above processes depict the method to calculate the contour $\left(x_{i},y_{i}\right)$ of the 1st segment. Other segments of the beam can be considered as cantilevers just like the 1st segment, while the initial bending angle of the local segment is the terminal bending angle of the previous segment. As illustrated in Figure 2b, $X_{n}O_{n}Y_{n}$ denotes the local coordinate system of the ith segment, whose initial bending angle is $\theta_{i-1}$ comparing with the global coordinate system.

Following the steps from Equation (Equation 4) to Equation (Equation 15), the coordinates $\left(x^{\prime },y^{\prime }\right)$ of the ith segment can be solved in the local coordinate system. Then, the transformation of transition and rotation is applied to obtain their coordinates in the global coordinate system as follows:(16)$$\left[\begin{array}{c} x_{i}\\ y_{i} \end{array}\right]=\left[\begin{array}{cc} \cos\theta_{i-1} & -\sin\theta_{i-1}\\ \sin\theta_{i-1} & \cos\theta_{i-1} \end{array}\right]\left[\begin{array}{c} x_{i}^{\prime }\\ y_{i}^{\prime } \end{array}\right]+\left[\begin{array}{c} x_{i-1}\\ y_{i-1} \end{array}\right]$$
where the θ and (x,y) with the subscript of ${}_{i-1}$ represent the bending angle and coordinates of the previous segment of the whole beam, respectively.

### 3.3. Stiffness Revising Based on Shell Bending Theory

As the above reconstruction results are calculated based on the beam hypothesis of the FFN plate, unavoidable errors are introduced to the bending stiffness (D=EI) that needs to be modified. Referring to the shell bending theory, the differential equations of the bending shell can be given as
(17)$$D\frac{d^{4}y}{d^{4}x}=q$$
(18)$$D=\frac{Eh^{3}}{12(1-\nu^{2})}$$
where the in-plane stretching/pressing force in *x* direction and longitudinal shear force in *z* direction are ignored. *q* represents the force per unit on the shell, and *D*, ν denote the bending stiffness and Poisson ratio of the shell, respectively.

Note that the bending stiffness EI of the beam is distinctive from the shell since the Poisson ratio needs to be considered. Thus, a coefficient β is introduced to revise the reconstruction results of shell structures with diverse width *b* and thickness *h*, resulting in
(19)$$D=\hat{E}I=\frac{\beta Eh^{3}}{12(1-\nu^{2})}$$
where β equals to
(20)$$\beta =\left\{\begin{array}{cc} 1/(1-\sqrt{1-h/b}) & \;\mathrm{for}\;h/b\leq 0.2\\ 1 & \;\mathrm{for}\;h/b>0.2 \end{array}\right.$$

After revising the bending stiffness of the shell model, the complete reconstruction algorithm of the FFN is given by Equation (Equation 15) with the parameter EI in Equation (Equation 13), replacing that depicted in Equation (Equation 19).

## 4. Evaluation of the Reconstruction Algorithm

Assisted by the FEM simulation tools, the modified SMRM are able to be evaluated and compared with the previous reconstruction methods.

### 4.1. Comparison with the Ko Theory

For the purpose of clarity, the structure in Figure 2a is taken as the research object whose structure parameters are depicted in Table 1

The ABAQUS software is introduced for numerical analysis of the FFN plate, while the geometric nonlinear solver is chosen and applied to FFN plate under several working cases listed in Table 2.

Figure 4a illustrates the reconstruction results of SMRM, KORM, as well as the FEM method that is exploited for comparison standards. The simulation results demonstrate that the SMRM and KORM are in line with FEM, which verifies the feasibility of our customized SMRM. Moreover, Figure 4b further depicts the reconstruction errors of the above two methods. The largest error of the KORM locates in case No. 3, reaching −0.066 mm, while the error of the SMRM is only −0.014 mm, which is 21.13% less than the error of the KORM. The result gives evidence that the SMRM is better than the KORM in accuracy.

### 4.2. Evaluation of the Modified Method

Considering the FFN plate as a shell, Figure 4c further illustrates the deformation simulation of the shell model whose length $l_{1}$, length $l_{2}$, and width *b* equal to 10 mm, 10 mm, and 10 mm, respectively. The shell models with diverse thicknesses *h* 2 mm, 1 mm, 0.5 mm, and 0.25 mm are chosen and calculated to avoid randomness, as the related h/b is 5, 10, 20, and 40, respectively. In ABAQUS, meshing is chosen as 0.2 mm based on the S4R shell unit, while geometric nonlinear is turned on in the compile setting. After applying forced displacement ($D_{1}=1$ mm, $D_{2}=3$ mm) to the two segments of the shell, the *y* coordinates of every unit on the shell are demonstrated in Figure 4c.

The above simulation results based on the FEM method are taken as the comparison standards, which is followed by the reconstruction of the deformed shell based on the SMRM method with/without modification. As shown in Figure 4d, the reconstruction errors of the modified SMRM are smaller than that of the unmodified SMRM. The maximum reconstruction error after modification could be reducing 10.78%, which verified the considerable accuracy and the improvement of the modified SMRM.

## 5. Results and Discussion

A prototype of the FFN plate has been made as the experimental platform that is illustrated in Figure 5a. The deformation of the FFN plate is achieved by four jacks mounted on the back where the flexible strain sensors locate as in Figure 5b,c. The strain sensors applied on the FFN plate are derived from our previous published paper [24,25].

Six kinds of contours that own the Mach number of 1.0, 1.3, 2.0, 2.5, and 3.0, respectively, have been utilized to evaluate the reconstruction method. Figure 6a depicts the output of the strain sensors from different positions and Mach numbers, which provides essential information for the strain distribution and reconstruction method. In order to obtain the reconstructed contour, the coordinate origin is chosen as the left edge in Figure 5a, and the *x*, *y*, and *z* directions are also shown in the figure. Note that a laser tracker measures the real contour of the FFN plate for comparison purposes. Following the steps of the customized SMRM, the distinctive reconstruction results of six Mach numbers are demonstrated in Figure 6b, where the contour data derived from the SMRM is in line with the experimental contour data under the error from −1 mm to 0.5 mm (Figure 6c). As illustrated from the experimental data, the maximum displacement located on the throat joint of 80 mm, which is eight times larger than the thickness of the plate. The above result guarantees the premise of the large deflection deformation problem.

Table 3 further depicts the reconstruction error of the FFN plate under different Mach numbers, as the maximum absolute error 3.56 mm is located on 1 Ma while the maximum relative error 3.97% is achieved on 1.5 Ma. The tendency of the absolute error and relative errors is drawn in Figure 6d, which demonstrates the maximum and minimum error to verify the primary reconstruction property of our developed SMRM. Note that the maximum displacement of the FFN plate’s edge is more essential for the deformation control instead of the root mean square error.

## 6. Experimental

Preparation of the flexible sensors: The flexible strain sensors based on graphene (Gr) and silver nanowires (AgNWs) were prepared and fabricated by the method reported in our previous paper [24]. The basic steps of the fabrication procedures were obtaining the AgNWs/Gr/AgNWs sensing film by vacuum filtration of their suspension, film transferring to flexible Polydimethylsiloxane (PDMS) substrate, and encapsulating the sensing film by another PDMS with liquid metal as electrodes. The graphene and AgNWs suspensions (4 mg/mL, water) were purchased from Nanjing XFNANO Materials Tech Co., Ltd., Nanjing, China The liquid metal is the alloy of Ga/In/Sn with a ratio of 68.5:21.5:10, and was bought from Zhenjiang Fan Yada Electronic Tech. Co., Ltd. PDMS (SYLGARD 184) was obtained from Dow Corning Co., Ltd.

Experimental platform: The FFN plate prototype that integrates the mechanical structures, motor devices, and control kits was obtained by fabricating the essential components and assembling the commercial products. The FFN plate was made from stainless steel (07Cr17Ni7Al, E = 203.4 GPa) by milling, with the thickness of 10 mm. The shape of FFN plate was driven by jacks connected with servo motors (IFX7042, SIEMENS AG Co., Germany) by software from SIEMES AG (SIMOUTION D410-2PN), and the position feedback was achieved by the grating ruler (LP40, TR Electronic GmbH). The flexible strain sensors were installed on the back of the FFN plate by partially cured PDMS, and the signals were obtained through liquid metal electrodes and silver wires (D = 0.04 mm, Kunshan Zhenyuhong New Material Co., Ltd., Shanghai, China).

Measurement: The data of the flexible strain sensors was measured by a resistance meter (0.1 μΩ∼1.2 GΩ, AT515, Changzhou Applent Instrument Co., Ltd., Changzhou, China). The contour of the FFN plate that acts as the standard was obtained by an absolute laser tracker (Leica AT960, Hexagon AB Co., Stockholm, Sweden)

## 7. Conclusions

This paper has taken a wind tunnel’s FFN plate as the research object to develop a highly accurate deformation reconstruction method for typical smart deformable structures. The strain–moment relationship of the FFN plate has been proposed by introducing the deformation theory of the equivalent beam, accompanied by the large deflection hypothesis. The differential equation and analytical solution of large deflection deformation of FFN plate have been derived based on Euler beam theory, which has been further modified by shell bending theory to increase accuracy. Subsequently, a FEM model of the FFN plate has been utilized to evaluate the developed SMRM compared with the classic KORM, resulting in the reconstruction error decreasing by 21.13%. Experimental results of different Mach numbers have been further obtained from the flexible strain sensors installed on a customized FFN prototype whose maximum reconstruction error by the SMRM method is only 3.97%. The reconstruction method proposed in this paper provides an efficient way for security monitoring and shape controlling of smart deformable structures.

## Figures and Tables

**Figure 1 micromachines-13-00910-f001:**
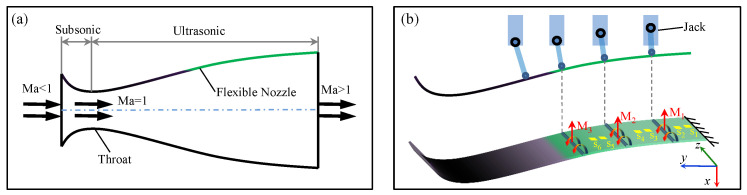
(**a**) Schematics of the FFN plate where Mach numbers vary from <1 to >1. (**b**) The deformation mechanism of FFN plate driven by four jacks.

**Figure 2 micromachines-13-00910-f002:**
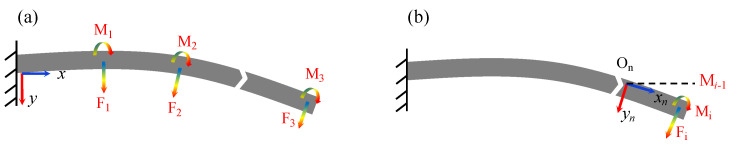
(**a**) The mechanical model of the equivalent beam. (**b**) The mechanism of reconstruction process of different segments.

**Figure 3 micromachines-13-00910-f003:**
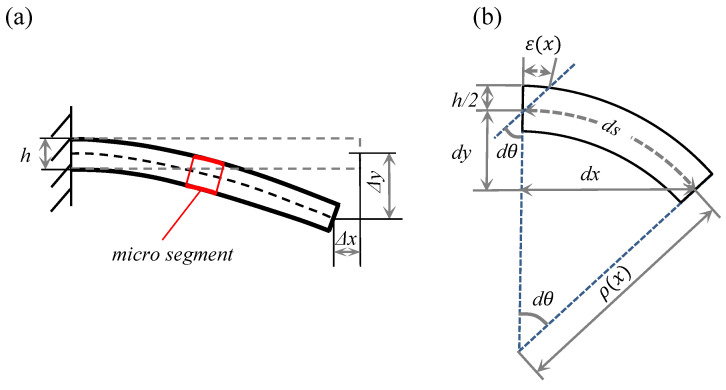
(**a**) The micro segment of the beam. (**b**) The relationship of the strain and curvature.

**Figure 4 micromachines-13-00910-f004:**
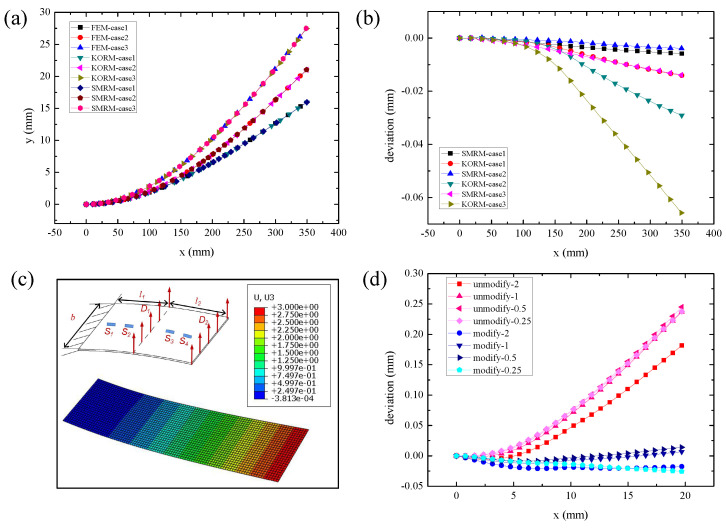
(**a**) Contrast of the reconstruction results of FEM, KORM, and SMRM. (**b**) The deviation of reconstruction of SMRM and KORM in three distinct cases. (**c**) The simulation results of the shell model. (**d**) Contrast of the reconstruction results of SMRM with and without modification.

**Figure 5 micromachines-13-00910-f005:**
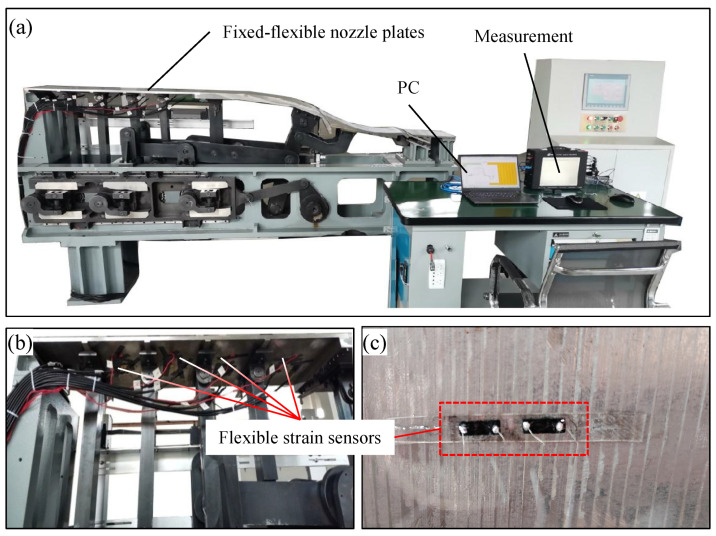
(**a**) The prototype of FFN plate acting as experimental platform. (**b**) The contribution of flexible strain sensors under the plate. (**c**) The flexible strain sensor.

**Figure 6 micromachines-13-00910-f006:**
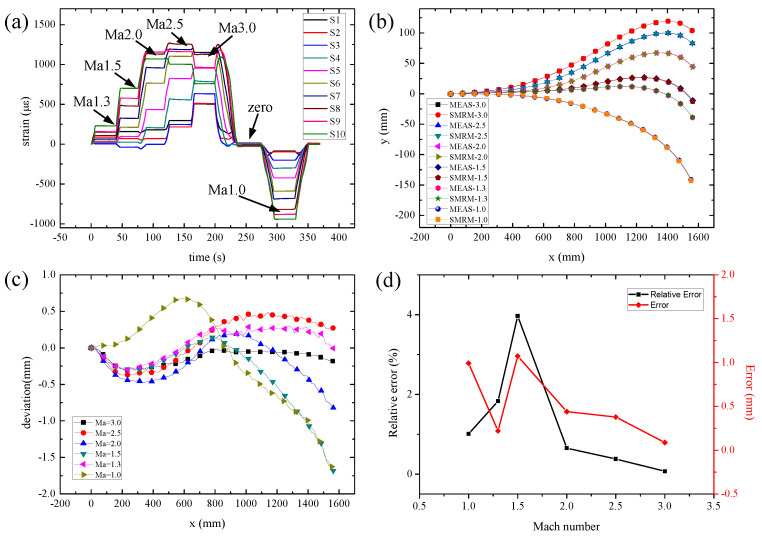
(**a**) Sensor outputs of different Mach numbers of the FFN plate. (**b**) The contrast of the reconstruction results by SMRM with experimental data. (**c**) The deviation of reconstruction results under different Mach numbers. (**d**) The relative error and absolute error of the FFN plate under different Mach number.

**Table 1 micromachines-13-00910-t001:** Structure parameters of the FFN plate model.

n	b (mm)	h (mm)	$l_{1}$ (mm)	$l_{2}$ (mm)	E (MPa)
2	5	5	175	175	$2\times 10^{5}$

**Table 2 micromachines-13-00910-t002:** Working cases for the method evaluation.

Condition	$F_{1}$ (N)	$F_{2}$ (N)	$M_{2}$ (N·mm)
1	5	10	20
2	−15	20	20
3	5	10	2000

**Table 3 micromachines-13-00910-t003:** The deformation reconstruction results of FFN plate under different Mach numbers.

Ma	Max. Displacement (mm)	Absolute Error	Relative Error
1.0	−98	0.99	1.01%
1.3	12	0.22	1.83%
1.5	27.03	1.07	3.97%
2.0	67.51	0.44	0.65%
2.5	99.98	0.38	0.38%
3.0	119.41	0.09	0.07%

## Data Availability

Experimental data is confidential due to the correlation with an ongoing research work.

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
