# Peer review of "A Highly Accurate Method for Deformation Reconstruction of Smart Deformable Structures Based on Flexible Strain Sensors"

_micromachines, 2022, doi:10.3390/mi13060910_

Round 1

Reviewer 1 Report

With the development of flexible and stretchable electronics, the flexible sensors have been widely used for safety monitoring. Due to its multifunctional requirements such as sensing and control, intelligent deformable structures are gaining more and more attention in the aerospace field, although there are still issues of relatively low accuracy and efficiency in the case of large deflection deformation. In this paper, a high-precision deformation reconstruction method based on sensing information from flexible strain sensors is developed for a fixed-flexible nozzle plate in a wind tunnel. The mechanical behavior of the FFN plate with large deflection deformation is modeled as a cantilever beam, and the relationship between its strain and moment is analyzed. In addition, the large deflection coefficients and shell bending theory are creatively used to derive and modify the strain-moment based reconstruction method, where the FFN plate profile is solved by a specific elliptic integral. Experimental results for different Mach numbers have been further obtained from flexible strain sensors mounted on a customized FFN prototype with a maximum reconstruction error of only 3.97%. The reconstruction method proposed in this paper provides an effective approach for the safety monitoring and shape control of smart deformable structures. The study is very interesting. However, the manuscript should be revised before it can be accepted.

1. In the figure caption of Figure 2, mechanics model should be mechanical model. Also in Figure 2, the rightmost F1 in a should be F3. The question mark in b should be M.

2. Lines 110-111 are incomplete, something should have been left out.

Author Response

1. In the figure caption of Figure 2, mechanics model should be mechanical model. Also in Figure 2, the rightmost F1 in a should be F3. The question mark in b should be M.

reply: Thank you for your attentive reading. We have corrected the spell and marks.

2. Lines 110-111 are incomplete, something should have been left out.

reply: We feel sorry about the extra sentence. We have removed some unnecessary words.

Reviewer 2 Report

This study developed a highly accurate deformation reconstruction method for smart deformable structures based on flexible strain sensors. Both the numerical and experimental approaches were applied to achieve the objective of this research. The authors concluded that the reconstruction method provides an efficient way for security monitoring and shape controlling of smart deformable structures.

This study is quite interesting. However, the manuscript is not well prepared. A scientific manuscript is suggested to be written using the IMRAD (introduction, methods, results, and discussion) structure. The authors did not completely comply with this writing structure. In fact, the authors used the following sections:

1. Introduction

2. The mechanics model of the FFN plate with multiple pivots

3. The derivation and modification of the reconstruction algorithm

4. Evaluation of the reconstruction algorithm

5. Results and discussion

6. Experimental

7. Conclusion

It would be better if the authors can consider the following writing suggestions.

1) Please divide the “Experimental” section into the method part and the results and discussion part.

2) Write the method part of the “Experimental” before the “Results and discussion” section.

3) Write the results and discussion part of the “Experimental” in the “Results and discussion” section.

In general, this study could provide the useful information and knowledge for readers. I recommended that this article could be published after revision.

Author Response

We appreciate your kind suggestions on our writing.  We can't agree more that IMRAD structure is a perfect method to prepare a scientific manuscript for most of the research work.  In this work, we would like to explain the processes of 'background, modeling, method, simulation, and results' to an unprofessional reader. Thus extra information has been moved to the 'Experimental section' in case of misleading the readers. We keep the 'Experimental section' for some professional readers to repeat our work. For the above reason, we kindly think the current structure is more suitable for the work.  

We really understand your concern about the clear description of our method and results. We have revised some sentences to emphasize the highlights of our manuscript.